# Acute Supplementation with Cannabidiol Does Not Attenuate Inflammation or Improve Measures of Performance following Strenuous Exercise

**DOI:** 10.3390/healthcare10061133

**Published:** 2022-06-17

**Authors:** Brett W. Crossland, B. Rhett Rigby, Anthony A. Duplanty, George A. King, Shanil Juma, Nicholas A. Levine, Cayla E. Clark, Kyndall P. Ramirez, Nicole L. Varone

**Affiliations:** Exercise Physiology Laboratory, School of Health Promotion and Kinesiology, Texas Woman’s University, Denton, TX 76204, USA; bcrossland@twu.edu (B.W.C.); aduplanty@twu.edu (A.A.D.); gking6@twu.edu (G.A.K.); sjuma@twu.edu (S.J.); nlevine@twu.edu (N.A.L.); cclark33@twu.edu (C.E.C.); kramirez29@twu.edu (K.P.R.); nvarone@twu.edu (N.L.V.)

**Keywords:** anti-inflammatory, athletes, cytokines, eccentric exercise, fatigue, hemp, muscle damage

## Abstract

Supplementation with cannabidiol (CBD) may expedite recovery when consumed after exercise. The purpose of this study was to determine if supplementation with CBD reduces inflammation and enhances performance following strenuous eccentric exercise in collegiate athletes. Twenty-four well-trained females (age = 21.2 ± 1.8 years, height = 166.4 ± 8 cm, weight = 64.9 ± 9.1 kg) completed 100 repetitions of unilateral eccentric leg extension to induce muscle damage. In this crossover design, participants were randomized to receive 5 mg/kg of CBD in pill form or a placebo 2 h prior to, immediately following, and 10 h following muscle damage. Blood was collected, and performance and fatigue were measured prior to, and 4 h, 24 h, and 48 h following the muscle damage. Approximately 28 days separated treatment administration to control for the menstrual cycle. No significant differences were observed between the treatments for inflammation, muscle damage, or subjective fatigue. Peak torque at 60°/s (*p* = 0.001) and peak isometric torque (*p* = 0.02) were significantly lower 24 h following muscle damage, but no difference in performance was observed between treatments at any timepoint. Cannabidiol supplementation was unable to reduce fatigue, limit inflammation, or restore performance in well-trained female athletes.

## 1. Introduction

The health benefits of exercise are well established through research, e.g., [1]. While exercise has numerous documented benefits, there is a period of time following exercise when individuals are unable to adequately perform due to fatigue [2]. Immediately following strenuous exercise, there is an observable drop in performance that can last from hours to several days depending on the intensity, duration, and type of exercise [3]. The underlying factors affecting this decrease in performance include myofibrillar disruption [4], swelling [5], reduced range-of-motion [6], an efflux of enzymes and proteins [7], and the inflammatory process [8]. These intercellular interactions can influence the time it takes to restore previous performance levels [9]. An inability to fully recover from strenuous bouts of exercise can increase the risk of injury and severely limit optimal performance [10].

While the exact time to fully regain performance capabilities cannot be precisely known for each athlete, a single muscle-damaging session typically results in a loss of strength and aerobic capabilities for up to four days [11,12]. For many, methods of optimal recovery from strenuous exercise and a rapid return to performance remain elusive. The primary goal of recovery from exercise for athletes and those who regularly engage in recreational exercise is aimed at a reduction in muscle soreness, and therefore a decreased time to the return of performance. Several methods to expedite the recovery from strenuous exercise have been documented. The methods that have shown some benefit are massage [13], stretching [14], water immersion [15], nutrition [16], and supplements [17].

The market for sports nutrition was estimated to be nearly $36 billion in 2020, with an expected average growth rate of 7.9% through 2028 [18]. A significant amount of research has been conducted in this area, with over 3500 articles published in 2021 alone [19]. Specifically, much of the focus has included nutritional products that are advertised to aid in the recovery from exercise, including supplementation with branched chain amino acids [20], milk-based products [21], and whey protein [22].

Recently, cannabidiol (CBD), which has been purported to aid in the recovery from strenuous exercise by decreasing inflammation and muscle damage [23], has gained popularity. The passing of the 2018 Farm Bill legalized the production and sale of hemp-derived cannabis products, which had previously been illegal under the Controlled Substances Act passed in 1970 [24]. Hemp, which is visually similar to marijuana, has a unique chemical content. Hemp contains less than 0.3% tetrahydrocannabinol (THC), the psychoactive component of the plant, while marijuana typically contains 10 to 15% THC [25]. The legalization of hemp allowed for the development and distribution of a large number of hemp-derived products. A significant increase in the production and distribution of CBD has been observed following the passing of the 2018 Farm Bill Act [26]. According to market research, CBD sales were over $600 million in 2018 and are expected to be more than $20 billion by 2022 [27].

Cannabidiol is one of over 100 cannabinoids found in the cannabis plant [28]. In hemp, CBD is the most abundant cannabinoid [29]. There are two primary forms of CBD supplements: isolate and full spectrum. Full spectrum CBD supplements contain all of the cannabinoids and terpenes naturally found in hemp [30]. Cannabidiol isolate supplements contain only CBD and trace amounts of other cannabinoids and terpenes [30]. Supplements containing CBD may be consumed via pills, lotions, tinctures, gum, powder, or edibles, with each delivery method resulting in variable bioavailability [31].

The therapeutic benefit of CBD supplementation may be useful with a variety of conditions, including those with epilepsy [32], Dravets syndrome [33], Lennox–Gastaut syndrome [34], and Alzheimer’s disease [35]. The positive effects of CBD are likely due to an ability to reduce neural inflammation, likely through reducing microglial cell migration and inhibiting nuclear factor kappa-light-chain-enhancer of activated B cells (NF-κB) signaling [36]. Indeed, 5 mg/kg of CBD was able to significantly reduce interleukin-6 (IL-6) concentrations [37], and reduce myocardial inflammation and oxidative stress [38], in mice models.

While the anti-inflammatory effects of CBD supplementation have been investigated across a number of different chronic conditions, there is currently very limited research investigating the effectiveness of CBD following intense exercise in humans. The purpose of this study is to determine if acute supplementation with CBD attenuates loss of performance, muscle damage, and inflammation following acute, strenuous, eccentric exercise in female collegiate athletes. With the results of this study, a better understanding of how CBD may influence muscle damage, performance, and inflammation may be gained. In addition, the results of this study will serve to inform consumers of CBD, coaches, trainers, and other healthcare professionals to their potential benefit in restoring markers of performance following strenuous exercise. The researchers hypothesize that acute supplementation with CBD will reduce inflammation, attenuate performance loss, and minimize muscle damage following intense eccentric exercise.

## 2. Materials and Methods

### 2.1. Participants

Twenty-seven female participants, ages 18 to 26 years, were recruited using a convenience sampling technique. An a priori power analysis was conducted using statistical software (G*Power 3.1.9.2, Dusseldorf, Germany) to determine the minimum sample size required to find statistical significance. With a desired power of 0.80, an alpha of level of 0.05, and a moderate effect size of 0.25 [39,40], it was determined that 24 participants would be required. Using word-of-mouth and email, participants were recruited from Texas Woman’s University and the University of North Texas. Participants were screened to include those who were current or previously NCAA Division I or II female athletes that had participated in a structured strength and conditioning program for a minimum of 6 months, were in the off-season portion of their competitive sport if current athletes, were deemed healthy as determined by a medical questionnaire and PAR-Q+, and had the ability to follow verbal directions. Participants were also screened to include those without injuries within the previous 30 days that limited the ability to train, orthopedic problems that could be exacerbated by exercise, known cardiovascular issues that may preclude them from exercise, consumption of any cannabis product or byproduct in the previous 30 days, consumption of anti-inflammatory medication for the previous 48 h, chronic exposure to, or consumption of, nicotine products within 48 h, and consumption of any alcohol within the 48 h prior to beginning the each trial. Throughout data collection, participants were asked to maintain normal dietary habits and avoid any significant changes in diet (e.g., starting a new diet, taking new supplements). Written consent was provided prior to participation, and all procedures were approved by the university’s institutional review board. Data collection began April 2021 and was completed in December 2021.

### 2.2. Study Protocol

The experimental study design for each treatment (CBD and placebo) is outlined in Figure 1. A double-blind, placebo-controlled, repeated-measures, crossover design was utilized to determine if differences existed among the participants with regards to markers of muscle damage, inflammation, and performance. Participants completed 6 visits in total, with visits 1 and 4 consisting of blood collection, performance measurements, and a muscle damaging protocol. Visits 2, 3, 5, and 6 involved blood collection and performance measures only. Following recruitment, researchers ensured each participant met the inclusion criteria. Once the informed consent had been completed, participants entered the study. Researchers randomized the treatment each participant received first, as well as the leg the participants used for each treatment. Once randomly assigned to a treatment group, participants notified the researchers upon onset of menses, at which point the first trial began within 7 days. A washout period of approximately 4 weeks was completed to allow for standardization of the menstrual phase and to ensure complete recovery from the muscle-damaging protocol. Participants were required to comply with all exclusion criteria in the period between visits 3 and 4.

#### 2.2.1. Visits 1 and 4: Muscle Damaging Sessions

For muscle damaging sessions, participants reported to the laboratory at approximately 07:00 for each session following an 8-h fast (water only, no caffeine) and had performed no strenuous physical activity for 72 h prior. Participants were measured for weight (kg) using a digital scale (Tanita BWB-800, Arlington Heights, IL, USA) and height (cm) using a stadiometer (Perspective Enterprises Model PE-AIM-101; visit 1 only). Following anthropometrics, a blood sample was collected. Participants then orally consumed the appropriate treatment, either 5 mg/kg of CBD or a placebo, and researchers noted the time in order to begin the muscle damaging session 2 h following consumption of the treatment. All participants consumed a standard snack following blood collection consisting of 4 kcal/kg of body weight (Clif Bar & Company, Emeryville, CA, USA; 250 kcal per 68 g serving; 18% fat, 68% carbohydrate, 14% protein). Participants were then free to leave the laboratory and reported back 2 h following consumption of the treatment, or at approximately 09:30. Participants were instructed to refrain from ingesting anything other than water or participating in physical activity outside of walking during this period.

##### Muscle Damaging Protocol

The muscle damaging exercise protocol consisted of 10 sets of 10 repetitions of eccentric leg extensions. A similar protocol elicits a sufficient muscle damaging and inflammatory response [41]. All repetitions were completed at an isokinetic speed of 30°/s on the dynamometer. Each repetition began at 0° of knee flexion and ended at 80° of knee flexion with a torque threshold of 406.7 N·m. For all repetitions, no concentric effort was exerted by the participants. Participants completed each set with no more than 3 s of rest between repetitions, with 1 min of complete rest given following each set. After completing the muscle damaging protocol, participants were free to leave the laboratory for 3.5 h and resume their normal daily nutritional habits.

#### 2.2.2. Visits 2, 3, 5, and 6: Performance Measure Sessions

Participants returned to the laboratory 24 and 48 h following each muscle damaging protocol in a fasted state (10 h, water only, no caffeine). Blood was collected upon arrival, and participants were supplied with the snack as they were in visits 1 and 4. Participants then began the same 10-min warm-up as the muscle damaging session. After completing the warm-up, static vertical jump was performed, and dynamic and isometric strength was assessed as outlined in the muscle damaging sessions. Approximately 28 days separated visit 3 with visit 4, with visit 4 occurring within the early follicular phase (days 2–7 after the start of menses).

##### Performance Measures

Upon returning to the lab, participants completed a 10-min warm-up at a self- selected cadence, maintaining heart rate under 60% of their age-predicted maximum (206.9 − (0.67 × age)) [42] on an electronically braked cycle ergometer (Ergomedic 828E; Monark Exercise AB, Vansbro, Sweden). After completing the 10-min warm-up, participants completed a countermovement vertical jump test utilizing a Just Jump mat (Probotics, Huntsville, AL, USA). All participants were allowed 5 submaximal warm-up vertical jumps with 30 s of rest between jumps prior to beginning the test. Participants stepped on the mat and placed their feet at hip width. Participants performed 3 maximum effort jumps with 1 min of rest between each repetition. The highest of the 3 maximum effort jumps was recorded to the nearest tenth of a cm.

All strength testing, and the muscle damaging protocol, was implemented using a Biodex System-3 isokinetic dynamometer (Biodex Medical Systems, Inc., Shirley, NY, USA). For all measures collected, the seat orientation was 90° and the seatback tilt was set to 80°. To familiarize themselves with the dynamometer, participants were allowed 5 repetitions at each speed at 50% of perceived maximal effort at each of the three tested speeds. In order to measure dynamic strength of the knee extensors, participants completed 10 maximal unilateral concentric-concentric repetitions at 60°, 180°, and 300°/s. The range-of-motion for the dynamic strength test started at 0° of knee flexion and ended at 90° of knee flexion. One minute of rest was given between each set, and peak torque (N·m) was recorded for the concentric knee extension at each speed for analysis.

After a 3-min recovery period, participants completed the isometric strength test on the dynamometer. Using the knee extensor of the leg chosen through random group assignment, participants produced a total of three maximal voluntary contractions for 5 s each, with one minute of rest between each repetition. For isometric strength testing, the knee angle was set at 90° of knee flexion with the lateral femoral epicondyle aligned with the axis of the dynamometer. The highest peak torque (N·m) exerted across the 3 trials was recorded for analysis. Participants completed a 5-min rest period before beginning the muscle damaging protocol.

Following the performance measures, participants completed the Visual Analogue Fatigue Scale (VAFS). The VAFS is a vertical 10-cm line with the top of the line representing “no fatigue”, while the bottom of the line represents “very severe fatigue”. Individuals were asked to draw a horizontal line across the vertical line indicating the level of fatigue they feel between “no fatigue” and “very severe fatigue.” The vertical distance was then measured between the top of the scale and the line drawn by participants in millimeters, and that number was recorded as the score.

#### 2.2.3. Blood Collection

Blood samples were collected at 2.5 h prior to, and 4, 24 and 48 h following the muscle damaging protocol for each treatment. Each blood sample, excluding the 4-h post muscle damaging sample, was performed following an 8 h fast (water only, no caffeine). For each blood sample, 10 mL of blood was collected via venipuncture from the antecubital vein in a normal clotting tube (SST II Advance, BD Biosciences, Haryana, Haryana, India). Arms were alternated for each successive blood draw. All samples were centrifuged after 30 min (1500× *g*, 10 min, 4 °C), and serum was stored at −80 °C until further analysis.

#### 2.2.4. Biochemical Analysis

Concentrations of IL-6, IL-10, and IL-1β were analyzed with the MILLIPLEX^®^ Map Human High Sensitivity T Cell assay using a custom bead panel kit (EMD Millipore Corporation, Bellerica, MA, USA). The analysis followed the manufacturer’s instructions for preparation and execution of testing. Concentrations of myoglobin (Mb) were analyzed with an Abnova MB (Human) ELISA kit (Abnova, Taipei, Taiwan). The manufacturer’s instructions were followed for all procedures, and all measures were performed in duplicate.

#### 2.2.5. Supplementation

Upon completion of the initial (−2.5 h) blood collection, the participants were supplied with 5 mg/kg of commercially available CBD isolate (verified 0% tetrahydrocannabinol (THC) via a certificate of analysis by the manufacturer) or a matched weight placebo (microcrystalline cellulose powder, Bulk Supplements, Henderson, NV, USA), orally via pill form. The same treatment was given to the participants immediately following completion of the muscle damaging protocol. Participants were then provided with the appropriate treatment and instructed to consume the supplement 10 h following the muscle damaging protocol. For each treatment, participants were asked to maintain regular nutritional habits until completion of the 48-h follow-up visit.

Both the placebo and CBD supplementation preparation were handled by the primary researcher. After treatments were prepared, they were provided to a secondary researcher who was responsible for randomizing treatment order and then dispensing the correct treatment to participants, as to blind the primary researcher from treatment administration. Both treatments were in powder form and put into identical Solaray vegetarian size 00 capsules (Nutraceutical, Salt Lake City, UT, USA). The placebo capsules were matched in weight to the treatment capsules.

### 2.3. Statistical Analysis

Statistics software (IBM SPSS Statistics v.24, Armonk, NY, USA) was used to analyze all data for performance, VAFS scores, and blood markers. First, a repeated-measures analysis of variance (RM ANOVA) was conducted to determine any differences in Mb concentrations and VAFS scores between groups and across the four timepoints (i.e., trials). Principle component analysis (PCA) was utilized to create a composite score at each trial for the 5 performance measures (i.e., concentric peak torque at 60°, 180°, and 300°/s, maximum peak isometric torque, maximum vertical jump height). Using dimension reduction through factor analysis, one composite score for each treatment and trial was created, which represented all measured performance variables. Based on the results of the PCA, a strong composite score could not be extracted, and therefore, a RM MANOVA was performed. Due to this procedure being omnibus, post hoc analyses were needed to determine significant differences in treatment, trial, and treatment/trial interaction. A univariate RM ANOVA was performed for all dependent variables after significant interactions were found with regards to trial. All significant interactions found between measured variables were analyzed using a post hoc Šídák test. A RM MANOVA was also used to examine potential differences in treatments with regard to inflammatory markers. Significance for all statistical analysis was set at 0.05.

## 3. Results

### 3.1. Participant Characteristics

Twenty-seven female NCAA athletes were recruited and signed a university- approved informed consent form. Twenty-four participants completed all procedures. Two participants withdrew due to relocation following graduation. Venipuncture was not possible on another participant, due to inaccessibility of vessels. All participants responded favorably to the treatment dose with no adverse reactions reported across all participants. Descriptive characteristics of participants who completed all required sessions of the study (*n* = 24) are outlined in Table 1.

### 3.2. Inflammatory Responses

In total, 86 of the 576 data points for inflammatory markers were not obtained at the time of analysis. Specifically, 26 data points were not obtained for IL-10, 6 data points were not obtained for IL-1β, and 54 data points were not obtained for IL-6. No significant differences (*p* = 0.573) were observed between the placebo and CBD treatments across the four measured trials. Additionally, no significant differences (*p* = 0.337) were observed between the four trials with both treatments combined. Concentrations of inflammatory markers over time can be seen in Figure 2.

### 3.3. Performance Measures

Based on the results of the PCA, a strong component (i.e., explained variance of composite score = 65%) could not be extrapolated, so a RM MANOVA was performed. No significant differences between the CBD and placebo treatments for dynamic strength at 60° (*p* = 0.479), 180° (*p* = 0.426), and 300°/s (*p* = 0.927), isometric strength (*p* = 0.671), and vertical jump (*p* = 0.806) were observed across all trials. Significant differences were found between trials (*p* = 0.034) with treatments combined. Therefore, a post hoc RM ANOVA was performed to determine which specific performance measures differed between the four trials.

Based on the results of the RM ANOVA with a post hoc Šídák adjustment, 60°/s peak torque and 5-s peak isometric torque were significantly lower than baseline (i.e., 2 h pre muscle damage) measures between trials. Compared to baseline measurements, 60°/s peak torque was significantly reduced at 4 h (*p* = 0.003) and 24 h (*p* = 0.001) following the muscle damaging session. Differences in 60°/s peak torque between trials can be seen in Figure 3a. With regards to 5-s peak isometric torque, performance was found to be significantly reduced at 4 h (*p* = 0.034) and 24 h (*p* = 0.023) following the muscle damaging session. Changes in 5-s peak torque between trials can be seen in Figure 3b.

### 3.4. Myoglobin

When both treatments were combined, a significant (*p* = 0.002) rise over baseline values in Mb levels was observed at the 4 h timepoint (see Figure 4). Based on the results of the RM ANOVA, no significant differences (*p* = 0.116) were observed between placebo and CBD treatments across all trials. Coefficient of variation values for Mb measures can be seen in Figure 4.

### 3.5. Visual Analog Fatigue Scale Scores

No significant differences (*p* = 0.126) for results on the VAFS were observed between the CBD and placebo treatments across all trials. Significant differences were observed between the four trials when treatments were combined. Following the baseline measurement, a significant (*p* < 0.001, < 0.001, = 0.01) increase in the VAFS score was observed at 4, 24, and 48 h, respectively, following muscle damage (See Figure 5).

## 4. Discussion

The purpose of this study was to investigate the effectiveness of CBD supplementation at reducing the markers of fatigue, loss of performance, and inflammation following intense eccentric exercise. Supplementation with CBD was unable to decrease markers of muscle damage, and subjective measures of fatigue, among collegiate athletes. Furthermore, a return of performance to baseline measures was not expedited, and the magnitude of inflammatory markers was similar to the placebo treatment. For individuals who exercise and athletes attempting to minimize fatigue associated with intense exercise, CBD supplementation does not appear to be an effective treatment.

### 4.1. Participants

Female collegiate athletes were targeted for recruitment in this study in an attempt to control for any acute training effects that may have occurred throughout the performance testing protocol, as well as differences in physiological response seen following exercise, in an untrained population. Female athletes were chosen in order to control for potential sex differences following intense exercise. In addition, the university in which the research was completed only allows for females to compete in athletics. Finally, it was cost prohibitive to include a similar number of age-matched males. In total, 24 participants completed all data collection requirements for the study. More specifically, female NCAA division 1 (*n* = 7) and division 2 (*n* = 17) athletes in the sports of volleyball (*n* = 2), basketball (*n* = 2), soccer (*n* = 8), gymnastics (*n* = 6), stunt (*n* = 1), and softball (*n* = 5) completed all procedures.

### 4.2. Study Protocol

In this study, 5 mg/kg of CBD was administered per dose to each participant. The amount and timing of the dosages was acute (i.e., hours) in nature, rather than chronic (i.e., weeks or months). A discussion of each of these will follow.

#### 4.2.1. Dose Amount

In rodent models, 5 mg/kg of CBD appears to be effective in reducing inflammation, and/or promoting an anti-inflammatory effect [37,43,44,45,46]. In humans, 150 mg of CBD oil taken immediately following, and 24 and 48 h following, muscle damage was unable to reduce subjective feelings of soreness or restore performance compared to a placebo treatment [47]. In a similar study, Isenmann et al. (2021) elicited muscle damage and supplied participants with either 60 mg of CBD or a placebo [48]. At the 72 h timepoint following muscle damage, serum measures of creatine kinase and Mb were significantly lower in those that consumed CBD [48]. Finally, Hatchett et al. (2020) found that 16.67 mg of CBD, when given with 1 mL medium-chain triglyceride (MCT) oil, was able to reduce subjective fatigue when compared to a placebo treatment [49]. To summarize, with regard to human participants, dosages in previous studies have been absolute, with 150 mg reported as the highest dose. We chose a relative dose of 5 mg/kg because of previous documented benefits in rodent models, and simply, we are not aware of the safety or tolerability of administering high dosages of CBD in humans. Based on the relative dose given in the study, the range of absolute dosages was 224 to 408 mg per timepoint administration, which was found to be tolerable for all participants.

#### 4.2.2. Timing of Dose

The timing of CBD administration, and the fact that it was acute in nature, is a ommon experimental setup with regard to athletes, performance, and supplement (with concurrent muscle damage) studies. Acute administration of CBD has been documented elsewhere [47,48]. Further, acute supplementation dosages and collection of health-related measures are common in studies that include muscle damage after eccentric exercise, e.g., [50]. Oftentimes athletes are asked to compete within a timeframe that does not allow for full recovery and are therefore at an increased risk for injury [51]. Acute supplementation of CBD was chosen for this study to determine if CBD could prove effective and decreasing fatigue following exercise, therefore providing significant practical application for athletes. Indeed, following a single soccer match, athletes can experience reduced performance and significant muscle damage for up to 72 h [52].

### 4.3. Cannabidiol Supplementation and Exercise

With regard to CBD supplementation, it is currently unclear what physiological mechanism would be responsible for changes in muscle damage, performance, or inflammation. It is hypothesized that the anti-inflammatory effects of CBD occur primarily through the nuclear factor-κB (NF-κB) pathway. Nuclear factor-κB is a group of transcription factors that play a large role in the body’s immune and inflammatory response [53]. Following intense exercise, NF-κB signaling is associated with the inflammatory response [54]. While no research has been conducted with human participants, CBD was able to reduce NF-κB signaling and decrease levels of inflammatory cytokines (TNF-ɑ, IL-6, and IL-1β), while increasing anti-inflammatory (IL-10) cytokine production, both in vitro and in vivo [55,56].

For the purposes of the current study, CBD isolate was chosen as the treatment due to the regulations within the NCAA and Texas state law, as well as the lack of research that includes supplements containing the phytocannabinoid CBD. Until recently, no published evidence on the effectiveness of CBD supplementation existed following exercise in human participants. With the increase in popularity of CBD supplementation and exercise, research on this topic is becoming more prevalent. Recently, 60 mg of CBD following exercise was found to attenuate creatine kinase and Mb response after 72 h following muscle damage via a back squat and depth jump [48]. It is unclear whether the CBD that was administered was isolate or full spectrum. After 24 and 48 h, no difference in creatine kinase and Mb levels were observed, which complements the results of the current study. Following muscle damage, serum Mb levels were measured to be 31.5 ng/mL, which is similar to the 32.77 ng/mL measured in the current study. An extended timeline may be needed to see a benefit with CBD supplementation with regard to markers of muscle damage.

In another recent study, supplementation with CBD oil was not able to reduce perceived soreness in untrained participants 24 and 48 h following a muscle damaging session [47]. Again, the composition of the CBD treatment (i.e., isolate vs. full-spectrum) was unclear, so it is difficult to relate to the current research. Similar to the current study, no significant differences were observed with VAFS scores between those that consumed CBD and a placebo at any time [47]. In summary, no significant benefit of CBD supplementation with regards to restoring performance loss following muscle damage has been documented [47,48], thus complementing the results of the current study.

### 4.4. Inflammation and Exercise

In the current study, supplementation with CBD was unable to mitigate the inflammatory or increase the anti-inflammatory response with several cytokines (IL-1β, IL-10, and IL-6) following 100 eccentric knee extension repetitions in 24 female collegiate athletes. The systemic inflammatory response following intense exercise begins with an influx of macrophages, neutrophils, and monocytes, stimulating the influx of cytokines [57]. The pro-inflammatory cytokine IL-1β is produced and released in a variety of cell types, and acts as an important mediator in the inflammatory response [58]. Following a marathon, IL-1β levels can be increased over 350% and remain elevated for more than five days [59]. Carbohydrate intake, antioxidant supplements, and cryotherapy may suppress IL-1β levels following the onset of stress [59,60,61]. In the current study, no significant (*p* = 0.573) differences in IL-1β between the CBD and placebo treatments across any of the four observed trials were observed.

Interleukin-6 levels can increase up to 100-fold following exercise, and the magnitude of IL-6 response is directly related to the intensity and duration of exercise, as well as the glycogen content of the muscle [62]. Released from the working muscle into circulation during exercise, IL-6 is primarily categorized as a pro-inflammatory cytokine, though it can stimulate the release of anti-inflammatory cytokines such as IL-10 and IL-1ra, and suppress concentrations of the pro-inflammatory cytokine TNF-ɑ [63]. Following intense eccentric exercise, a rise in IL-6 is significantly correlated to the subsequent rise in serum markers of muscle damage (i.e., creatine kinase, Mb, lactate dehydrogenase) [64]. With this broad spectrum of mechanisms of action, regulation of IL-6 could play an important role in an individual’s ability to achieve a high-performance state following strenuous exercise. Interleukin-6 concentrations following strenuous, eccentric exercise can vary. For example, Willoughby et al. (2003) observed a three-fold increase following 70 repetitions of eccentric exercise (i.e., back squat) [41]. Smith et al. (2000) reported similar results following 48 eccentric repetitions of leg curl and bench press exercises [65]. However, Wilborn et al. (2017) observed no significant rise in IL-6 levels 24 h following 240 eccentric leg extension reps [66]. Similarly, Cornish and Johnson (2014) reported no changes in IL-6 concentrations following 60 repetitions of eccentric leg extension [67]. In the current study, serum levels of IL-6 in female collegiate athletes did not significantly differ between those that consumed CBD or a placebo or across multiple timepoints following 100 repetitions of eccentric leg extensions. Additionally, there were no significant differences between trials, indicating the muscle damaging stimulus may not have been demanding enough to elicit an inflammatory response as seen in studies using a similar protocol. These results, in addition to IL-1β concentrations being unchanged, suggest that supplementation with CBD may not have the ability to limit the inflammatory response following exercise.

Cannabidiol has been marketed and sold as containing anti-inflammatory ingredients and is thus helpful when it is consumed following exercise. When given 5 mg/kg of CBD isolate 2 h prior to, immediately after, and 10 h after intense eccentric exercise, there was no significant (*p* = 0.573) increase in the anti-inflammatory cytokine IL-10 when compared to a placebo. Interleukin-10 is a cytokine that interacts with immune and nonimmune cells and acts to mitigate the inflammatory response following stress [68]. The primary function of IL-10 is to regulate and ultimately stop the inflammatory response across several cell types, and it may also enhance myogenesis [69,70]. There is no agreement as to the nature of response of IL-10 following intense eccentric exercise. Hirose et al. (2004) observed a two-fold increase in IL-10 following 30 repetitions of eccentric elbow flexion [71]. However, Cornish and Johnson (2014) reported undetectable IL-10 levels following 60 eccentric knee extensions [67]. Several factors, including sex, training status, and genetic predisposition, may affect the inflammatory response to exercise, which may help explain the variation in research findings. Given the inability of CBD supplementation to reduce serum markers of the pro-inflammatory cytokines IL-1β and IL-6 or increase serum markers of the anti-inflammatory cytokine IL-10 following intense eccentric exercise, CBD may not reduce inflammation, as purported by numerous marketing strategies for the supplement.

### 4.5. Performance

It is not uncommon for professional, collegiate, and even amateur athletes to be asked to complete incredibly demanding maximal-effort competitive sessions on less than 48 h rest. The ability to reduce fatigue and minimize performance loss between these sessions can play a vital role in subsequent performances. Perhaps more importantly, injury rates are higher when athletes are asked to perform competitive efforts on short (24 to 48 h) rest [51]. Moreover, training status is an important factor in an individual’s ability to recover from exercise, with lesser trained athletes possessing a slower return to a resting state with regard to inflammation, force production, autonomic nervous system activity, soreness, and performance [72]. Participants in the current study were well-trained, with each participant competing at their respective sport for a minimum of six years.

Compared to 60°/s peak torque and 5-s isometric peak torque, the remaining measured performance variables (i.e., peak torque at 180 and 300°/s and vertical jump) require less sustained maximal contraction and are rather explosive in nature. A significant loss in performance across both treatments at 4 and 24 h following muscle damage with regards to 60°/s peak torque (*p* = 0.003 and 0.001, respectively) and 5-s isometric peak torque (*p* = 0.034 and 0.023, respectively) was observed in this study. This was expected, as it is known that following muscle damage, sustained maximal contractile strength is among the first physiological variables to decline, while high speed submaximal contractile properties remain stable [73]. Central fatigue is a likely contributor to this observation, as the decrease in sustained maximal contractile strength is due to impaired muscle function and the ability of the central nervous system to optimally activate the muscle [74].

### 4.6. Muscle Damage and Fatigue

There is significant debate with regard to the identification and quantification of exercise-induced muscle damage (EIMD) [75]. Several methods of determining the severity of EIMD include the use of blood marker concentrations, MRI, muscle biopsy, subjective soreness scales, and girth measurements [76]. Of these methods, change in blood serum muscle proteins is a commonly used and accepted method of determining the scope of EIMD. Serum levels of creatine kinase, Mb, and lactate dehydrogenase are the most commonly used proteins in identifying EIMD [77]. For this research study, Mb was chosen due to the short time to peak (i.e., 24 h), and faster return to baseline, compared to other muscle proteins [77].

Supplementation (i.e., BCAA’s, omega 3’s, antioxidants, carbohydrate/protein mixes) that aims to reduce EIMD can yield varying results [19,78,79]. Supplementation with CBD in the current study was given 2 h prior to the muscle damaging protocol allowing for ample time to reach max concentrations in the system [31]. No significant differences (*p* = 0.116) were observed with EIMD, as indicated by Mb levels between those that consumed CBD or a placebo. Peak concentration of CBD can be reached between 1.5 and 3 h following oral ingestion, indicating that CBD levels were at or near peak concentrations during the muscle damaging session [31]. Based on our results, CBD was not effective at reducing EIMD following intense eccentric exercise. Therefore, it does not appear that CBD supplementation at an oral dosage of 5 mg/kg is able to attenuate Mb response following intense eccentric exercise.

An individual’s prior training history and current training status play a large role in the magnitude and duration of EIMD following strenuous exercise [80]. The time required for Mb levels to return to baseline is increased for untrained individuals when compared to trained individuals. In this study, a significant (*p* = 0.002) rise in Mb levels 4 h after participants performed 100 repetitions of eccentric knee extensions was recorded, with Mb levels returning to near baseline after 24 h. In untrained participants, Mb levels remained significantly increased after 48 h using a similar protocol [81,82]. The results of the current study are similar to previous studies in which Mb returned to baseline 24 h following intense exercise in trained athletes [83,84]. This is significant, as the magnitude of Mb response is associated with the mode, intensity, and duration of the exercise performed [48,85,86,87].

In the current study, subjective fatigue at 4, 24, and 48 h after the muscle-damaging session were all significantly (*p* < 0.001, <0.001, =0.01, respectively) elevated over baseline. Participants in this study perceived fatigue with both treatments and were thus less prepared to perform at an optimal level up to 48 h following the muscle-damaging session. Previous researchers have hypothesized that the anxiolytic effects of CBD may positively influence VAFS scores [88], and a significant benefit has been observed using full spectrum CBD and medium-chain triglyceride (MCT) oil [49]. In the current study, CBD was not able to significantly (*p* = 0.126) reduce VAFS scores, indicating that CBD supplementation is not effective in reducing subjective measures of fatigue and EIMD.

### 4.7. Limitations

There were several limitations to this study. The recruitment of female collegiate athletes exclusively may have affected the outcome of the study. Differences exist with regards to the hormonal and inflammatory response to exercise based on age, sex, and training status, and thus the results of this study may only be presumed accurate for well-trained females between the ages of 18 and 26 years [89].

The specific CBD product and dosage chosen may also be a limitation. Cannabidiol was administered in pill form, which has a slower absorption rate and lower peak concentration when compared to CBD administered via oils, inhalation, or injection [31]. Due to these factors, a higher dosage was needed for oral pill administration. Although carefully selected after a thorough review of the literature, the selected timing dosage (three doses of 5 mg/kg) of CBD may also be a potential limitation.

The inflammatory cytokines IL-1β, IL-10, and IL-6 were analyzed using a Luminex MagPix^®^ system. While this method provides efficiency via multispec analysis (i.e., multiple targets analyzed per well), there are potential errors in detection due to intraplate variance of the magnetic beads. In other cases, samples may be out of the detectable range. Because of these factors, we were forced to omit several samples, due to either low detectability in the sample or errors in the analysis of magnetic beads. This is a limitation of the study because it was not financially feasible to purchase the required supplies to remeasure samples.

Finally, diet was not controlled. It is well known that diet affects body composition and inflammation, thus potentially affecting the outcomes in this study. Participants in this study were instructed not to make any conscious changes to their regular dietary intake or use of dietary supplements during the intervention.

## 5. Conclusions

The ability to improve performance through enhancing recovery has recently gained in popularity. Modalities such as self-myofascial release, compression garments, massage, and cold-water immersion may expedite the recovery process following exercise [90]. Nutritional interventions, including BCAA’s, nitrates, and β-hydroxy β-methylbutyrate, may also be of benefit [19]. The purpose of this study was to determine if CBD supplementation reduces fatigue and inflammation, and enhances performance, following eccentric exercise. Cannabidiol was unable to lower subjective fatigue scores, mitigate muscle damage, expedite a return of performance, or limit the inflammatory response in female collegiate athletes. Cannabidiol therefore is not recommended as an effective addition to an individual’s recovery protocol. It is recommended that further research with varying CBD supplements be conducted in order to determine if other phytochemicals found in the cannabis plant prove effective as a means of expediting recovery.

## Figures and Tables

**Figure 1 healthcare-10-01133-f001:**
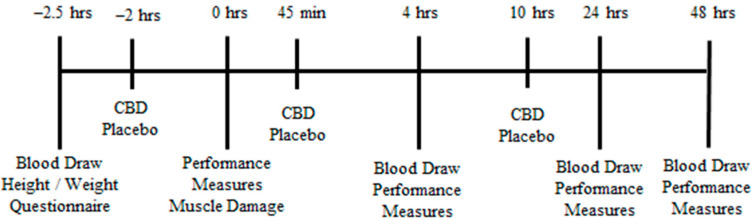
Study timeline for a given treatment. CBD = Cannabidiol; hrs = hours.

**Figure 2 healthcare-10-01133-f002:**
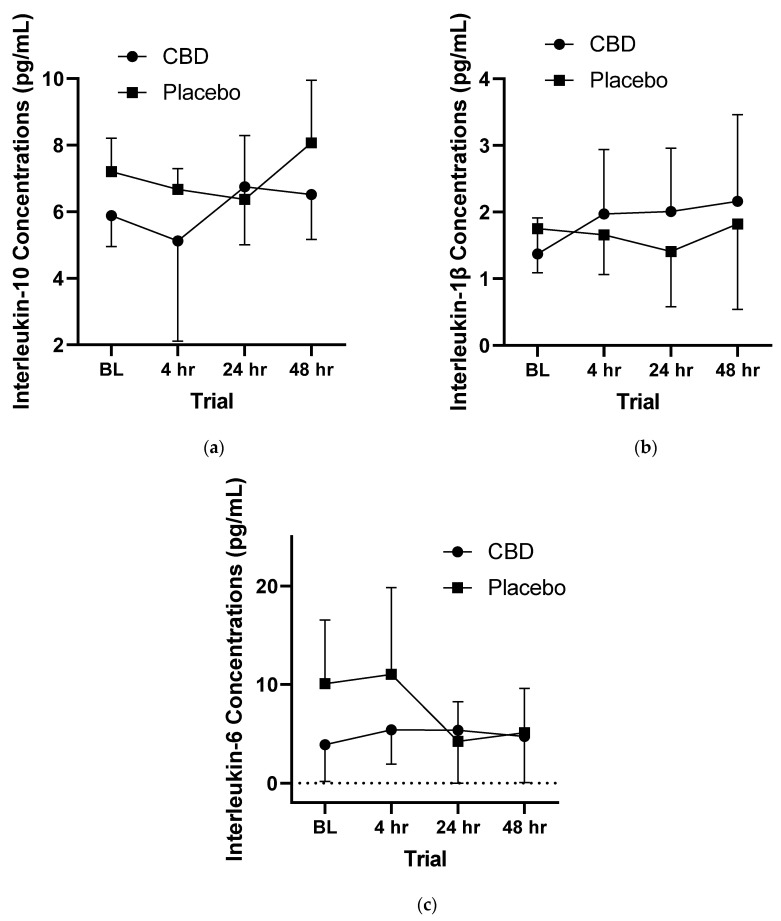
Concentrations (pg/mL) of (**a**) interleukin-10, (**b**) interleukin-1β, and (**c**) interleukin-6 at various timepoints. BL = 2 h prior to muscle damaging protocol, 4 hr = 4 h post-muscle damaging protocol, 24 hr = 24 h post-muscle damaging protocol, 48 hr = 48 h post-muscle damaging protocol. (*n* = 24).

**Figure 3 healthcare-10-01133-f003:**
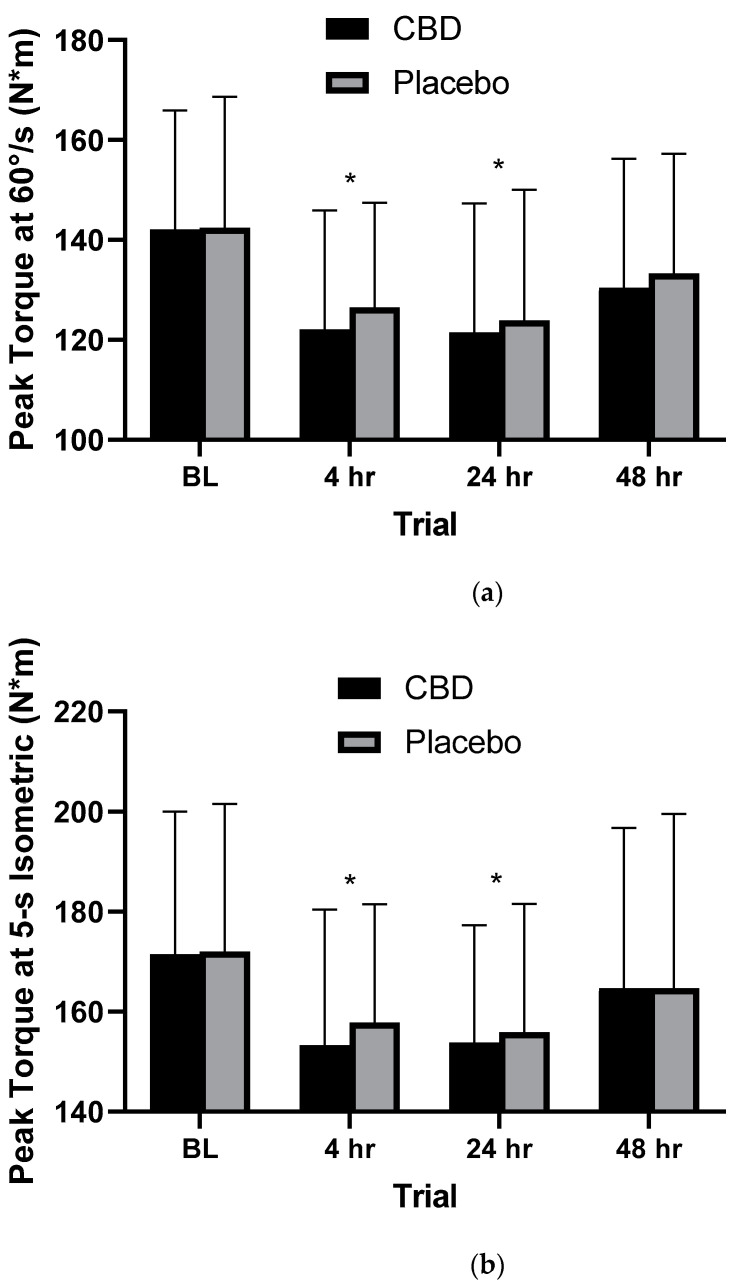
(**a**) Peak torque at 60°/s and (**b**) 5-s peak torque measures at various time points. * Statistically significant difference (*p* < 0.05) observed from baseline measurement. BL = 2 h prior to muscle damaging protocol, 4 hr = 4 h post-muscle damaging protocol, 24 hr = 24 h post-muscle damaging protocol, 48 hr = 48 h post-muscle damaging protocol. (*n* = 24).

**Figure 4 healthcare-10-01133-f004:**
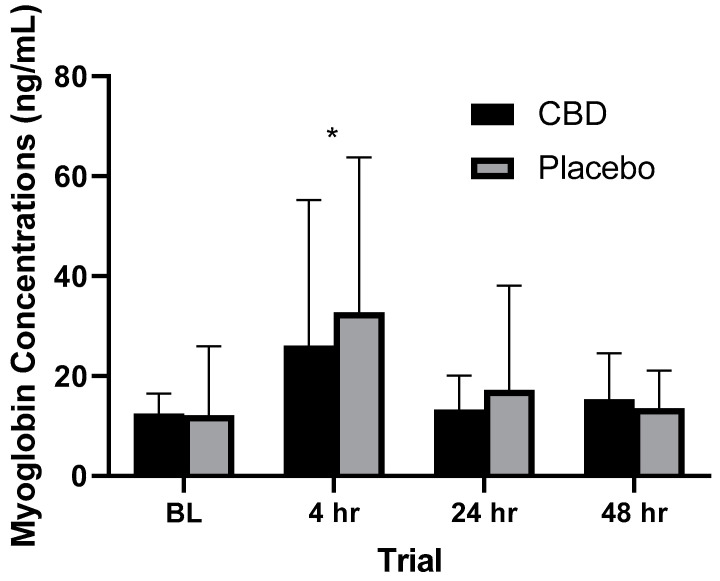
Myoglobin values at various time points. * Statistically significant difference (*p* = 0.002) observed from baseline measurement. BL = 2 h prior to muscle damaging protocol, 4 hr = 4 h post-muscle damaging protocol, 24 hr = 24 h post-muscle damaging protocol, 48 hr = 48 h post-muscle damaging protocol. (*n* = 24).

**Figure 5 healthcare-10-01133-f005:**
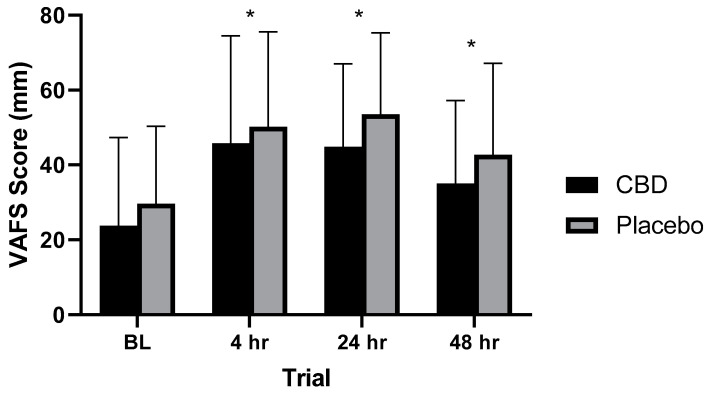
Visual Analog Fatigue Scale scores at various time points. * Statistically significant difference (*p* < 0.05) observed from baseline measurement. BL = 2 h prior to muscle damaging protocol, 4 hr = 4 h post-muscle damaging protocol, 24 hr = 24 h post-muscle damaging protocol, 48 hr = 48 h post-muscle damaging protocol. (*n* = 24).

**Table 1 healthcare-10-01133-t001:** Participant characteristics (*n* = 24).

Age (years)	21.2 ± 1.8
Height (cm)	166.4 ± 8.0
Weight (kg)	64.9 ± 9.1
BMI (kg/m^2^)	23.7 ± 2.4
Vertical Jump Performance (cm)	49.3 ± 8.1

Note. Values are presented as mean ± SD; BMI = body mass index, calculated as body mass (kg)/height (m^2^).

## Data Availability

Not applicable.

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
