# Peer review of "Acute Supplementation with Cannabidiol Does Not Attenuate Inflammation or Improve Measures of Performance following Strenuous Exercise"

_healthcare, 2022, doi:10.3390/healthcare10061133_

Round 1
Reviewer 1 Report
The study is interesting, but has numerous limitations. Some of them are considered by the authors but, in any case, they do not make the results easily interpretable. Such limitations reduce the importance of the study.
1. From the data in the literature used as a reference, the CBD dosage was intraperitoneal. In the study, however, the authors administer a pill, so the concentration in the blood is not the same
2. The authors did not evaluate TNF-alpha. that value would have been useful.
3. From the evaluations on interleukins they lose a lot of data. This makes the available data less reliable.
4. the conclusions seem forced. They are based on a study carried out with different concentrations (ref n. 37)
Author Response
Reviewer 1
Thank you for your comments. To distinguish between you and the other reviewers, all answers to your comments and any changes in the manuscript have been highlighted in purple.
From the data in the literature used as a reference, the CBD dosage was intraperitoneal. In the study, however, the authors administer a pill, so the concentration in the blood is not the same.
In choosing the specific CBD supplementation, it was difficult to find isolate that contained no other additives that could potentially have an effect on inflammation or other measures. Pill form was chosen due to this. Since pill form results in lower concentration rates a relatively large (5 mg/kg) dose was chosen at 3 different timepoints to ensure concentrations in the system were elevated.
The authors did not evaluate TNF-alpha. that value would have been useful.
TNF-alpha measurement was discussed during conception of the study but was omitted due to financial limits as well as some research using similar methodology not finding a significant rise in TNF-alpha following eccentric exercise.
Tanabe, Y., Maeda, S., Akazawa, N., Zempo-Miyaki, A., Choi, Y., Ra, S. G.,.. & Nosaka, K. (2015). Attenuation of indirect markers of eccentric exercise-induced muscle damage by curcumin. European journal of applied physiology, 115(9), 1949-1957.
Clifford, T., Bell, O., West, D. J., Howatson, G., & Stevenson, E. J. (2016). The effects of beetroot juice supplementation on indices of muscle damage following eccentric exercise. European journal of applied physiology, 116(2), 353-362.
From the evaluations on interleukins they lose a lot of data. This makes the available data less reliable.
We recognize this issue and have discussed the matter in the limitations section. While there was some missing data, the authors feel the inflammatory cytokine measures were an integral part of the study and provided relevant information.
The conclusions seem forced. They are based on a study carried out with different concentrations (ref n. 37)
Reference number 37 is not found in the Conclusions section, only in the Introduction section. The authors have summarized the key takeaways from the study, and provide a recommendation based on the results of the study.

Reviewer 2 Report
The aim of this study was to determine if supplementation with CBD reduces inflammation and enhances performance following strenuous eccentric exercise in collegiate athletes. This is a well-written manuscript presenting novel findings. However, I have a few minor comments.
In the Introduction section, the authors should explain the reasons of testing CBD on female athletes. An explanation is needed.
Also, the authors should add a hypothesis.
The authors wrote: " With a desired level of power at .80, an alpha (É‘) level at .05, and a moderate effect size of .25 (f), it was determined that 24 participants would be required." Why the authors chose a moderate ES? I think that they should use a comparable study even on male athletes to calculate the required sample size.
The calculation of the required sample size should be removed to the Participants subsection.
There are several limitations, but I think the manuscript could be accepted after responding adequately to a few minor comments.
Author Response
Reviewer 2
Thank you for your comments. To distinguish between you and the other reviewers, all answers to your comments and any changes in the manuscript have been highlighted in red.
In the Introduction section, the authors should explain the reasons of testing CBD on female athletes. An explanation is needed.
We have addressed this comment, but decided it would be best to discuss this in the Discussion section (section 4.1. Participants). Thus, this has been added, lines 351-354. Female athletes were chosen in order to control for potential sex differences following intense exercise. In addition, the athlete participant population at the university where the research was conducted is only female. Finally, it was cost prohibitive to age (and sport) match with a group of males.
Also, the authors should add a hypothesis.
We have addressed this comment at the end of the Introduction section, lines 91-93. The researchers hypothesize that acute supplementation with CBD will reduce inflammation, attenuate performance loss, and minimize muscle damage following intense eccentric exercise.
The authors wrote: " With a desired level of power at .80, an alpha (É‘) level at .05, and a moderate effect size of .25 (f), it was determined that 24 participants would be required." Why the authors chose a moderate ES? I think that they should use a comparable study even on male athletes to calculate the required sample size.
Thank you for your comment. We chose a moderate effect size based on three comparable studies: 1) a large effect size was observed with 11 male participants using a similar timing and training protocol, albeit with betalain-rich concentrate supplementation (Vitti et al., 2020); 2) a moderate to large effect size was observed with 17 males using a similar timing and training protocol, albeit with curcumin supplementation (Nicol et al., 2015); 3) significant differences were found when exploring the relationship between muscle damage and heart rate variability in 6 female collegiate athletes in a pilot study in our laboratory (Crossland et al., 2018).
The article references have been added to the manuscript in line 101.
References
Crossland, B., Sokoloski, M., & Rigby, B. R. (2018). The relationship between heart rate variability and skeletal muscle damage in female collegiate athletes. International Journal of Exercise Science Conference Abstracts, 2(10).
Nicol, L., Rowlands, D., Fazakerly, R., & Kellett, J. (2015). Curcumin supplementation likely attenuates delayed onset muscle soreness (DOMS). European Journal of Applied Physiology, 115(8), 1769-1777.
Vitti, S., Bruneau, M., Leyshon, K., Sotir, S., Headley, S., & O’Neill, E. (2021). The effects of betalain-rich concentrate supplementation in attenuating muscle damage following eccentric exercise. Journal of Human Sport and Exercise, 16(1), 112-121.
The calculation of the required sample size should be removed to the Participants subsection.
This has been moved to lines 98-102.
There are several limitations, but I think the manuscript could be accepted after responding adequately to a few minor comments.
Thank you for your comment.

Reviewer 3 Report
The authors provide convincing support for the value of this investigation by thoroughly describing the use of CBD in treating multiple inflammatory processes. It makes sense to test CBD's effect in post-exercise recovery.
Lines 49-51; Are more recent data available (2016, 2017 are rather old).
Line 59: aesthetics refers to beauty; ;use "visually' instead.
91: When was the study conducted?
94, 100: State the age range only once.
The lack of dietary and dietary supplement assessment needs to be explained. Diet is an important factor in inflammation and performance.
Line 274: Figure numbering is not cited correctly.
Figures: In the footnote indicate the number of participants for each panel.
The discussion is well referenced and answered many of my questions.
Author Response
Reviewer 3
Thank you for your comments. To distinguish between you and the other reviewers, all answers to your comments and any changes in the manuscript have been highlighted in blue.
Lines 49-51; Are more recent data available (2016, 2017 are rather old).
The text has been updated in lines 49-52 to read: “The market for sports nutrition was estimated to be nearly $36 billion in 2020, with an expected average growth rate of 7.9% through 2028 [18]. A significant amount of research has been conducted in this area, with over 3,500 articles published in 2021 alone [19].” References 18 and 19 have been updated in the References section.
Line 59: aesthetics refers to beauty; use "visually' instead.
This has been updated in line 59.
91: When was the study conducted?
This has been added in lines 120-121: “Data collection began April 2021 and was completed in December 2021.”
94, 100: State the age range only once.
The age range originally in line 100 has been removed.
The lack of dietary and dietary supplement assessment needs to be explained. Diet is an important factor in inflammation and performance.
Information been added in line 115-117: “Throughout data collection, participants were asked to maintain normal dietary habits and avoid any significant changes in diet (e.g., starting a new diet, taking new supplements).” With this directive, and the consideration of all inclusion criteria, the authors feel that this has adequately been addressed.
Line 274: Figure numbering is not cited correctly.
Figure numbering has been updated in lines 280, 303, 306, 316, and 325
Figures: In the footnote indicate the number of participants for each panel.
Figures have been updated in lines 288, 313, 330, and 335.
The discussion is well referenced and answered many of my questions.
Thank you for your comment.

Reviewer 4 Report
Crossland et al. present an interesting study on the effectiveness of full-spectrum CBD oil on muscle recovery. Overall, the paper is well-written and will contribute nicely to the current literature on a fascinating topic.
Major concern:
1. It is unclear why the set dose and duration were used in the present study.
Two previous investigations (Cochrane-Snyman, 2021 & Hatchett, 2020 [https://www.kheljournal.com/archives/2020/vol7issue2/PartB/7-2-4-412.pdf]) used a similar study design and supplementation procedure. While the introduction set up the reason for the use of "full-spectrum" vs isolate CBD rather well, it is unclear why another acute trial was conducted. The manuscript may be improved by a thorough discussion of the reason for the use of 5 mg/kg and an acute approach. The effects attributed to CBD are related to anti-inflammation; however, tissue CBD incorporation may have effects related to muscle recovery. See recent CBD dosing paper (albeit in mice), https://www.mdpi.com/2072-6643/14/10/2101/htm.
Minor recommendations:
1. It may be appropriate to include "Acute supplementation" in the title.
2. Considering adding the Hatchett (2020) paper to the discussion.
Author Response
Reviewer 4
Thank you for your comments. To distinguish between you and the other reviewers, all answers to your comments and any changes in the manuscript have been highlighted in green.
It is unclear why the set dose and duration were used in the present study.
Two previous investigations (Cochrane-Snyman, 2021 & Hatchett, 2020 [https://www.kheljournal.com/archives/2020/vol7issue2/PartB/7-2-4-412.pdf]) used a similar study design and supplementation procedure. While the introduction set up the reason for the use of "full-spectrum" vs isolate CBD rather well, it is unclear why another acute trial was conducted. The manuscript may be improved by a thorough discussion of the reason for the use of 5 mg/kg and an acute approach. The effects attributed to CBD are related to anti-inflammation; however, tissue CBD incorporation may have effects related to muscle recovery. See recent CBD dosing paper (albeit in mice), https://www.mdpi.com/2072-6643/14/10/2101/htm.
Thank you for your comment. We would like to address the dosage of 5 mg/kg first. In rodent models, 5 mg/kg of CBD appears to be effective in reducing inflammation, and/or promoting an anti-inflammatory effect. Specifically, Durst et al. found that 5 mg/kg of CBD was able to significantly reduce infarct size, leukocyte count, and interleukin-6 (IL-6) concentrations in mice. Malfait et al. (2000) found that 5 mg/kg elicited an anti-inflammatory effect in rodents with collagen-induced arthritis. Borrelli et al. (2009) investigated the protective effects of CBD in a mouse model of colitis. Mice were given 5 mg/kg of CBD, and concentrations of the pro-inflammatory IL-1b were significantly reduced, while concentrations of the anti-inflammatory IL-10 were increased. Five mg/kg of CBD was able to significantly reduce the activation of vascular cell adhesion protein 1 (VCAM-1) both in vitro and in vivo. This signaling protein is activated by pro-inflammatory cytokines (Mecha et al., 2013). Finally, 5 mg/kg of CBD reduced inflammation by 48% in mice models (Gallily et al., 2018). In human trials, there have been 3 known studies conducted with CBD administration and exercise. In a study investigating subjective soreness measures and performance, researchers found that 150 mg of CBD oil taken immediately following, and 24 and 48 hrs following, muscle damage was unable to reduce subjective feelings of soreness or restore performance compared to a placebo treatment (Cochrane-Snyman, et al., 2021). In a similar study, Isenmann et al. (2021) elicited muscle damage, and supplied participants with either 60 mg of CBD or a placebo immediately after. Blood markers of muscle damage were measured, at 24, 48, and 72 hrs. Cannabidiol supplementation was given immediately following the muscle damaging protocol and was administered via a 60 mg solubilisat with water. No significant differences between treatment groups at the 24 and 48 hr timepoints for any of the measured variables were observed. However, at the 72 hr timepoint, serum measures of creatine kinase and Mb were significantly lower in the CBD group, indicating a reduction in blood markers of muscle damage. Finally, Hatchett et al. (2020) found that 16.67 mg of CBD, when given with 1 ml medium-chain triglyceride (MCT) oil, was able to reduce subjective fatigue when compared to a placebo treatment. In summary, it appears that a dosage of 5 mg/kg of CBD in rodent models of disease is effective in attenuating inflammation. In humans, dosages in previous studies have been absolute, with the highest being 150 mg. We chose a relative dose of 5 mg/kg because of previous documented benefits in rodent models, and simply, we are not aware of the safety/tolerability of administering high dosages of CBD in humans. Based on the relative dose given in the study, the range of absolute dosages was 224 to 408 mg per timepoint administration, which was tolerable.
To address the acute administration issue, this is a common experimental setup with regard to athletes, performance, and supplement (with concurrent muscle damage) studies. As noted above, both Cochrane-Snyman et al. and Isenmann et al. conducted acute administration studies. Acute supplementation dosages and collection of health-related measures are common in studies that include muscle damage after eccentric exercise (e.g., Kim and So, 2019). Oftentimes athletes are asked to compete within a timeframe that does not allow for full recovery, and are therefore at an increased risk for injury (Mason et al., 2022). Acute supplementation of CBD was chosen for this study to determine if CBD could prove effective and decreasing fatigue following exercise, therefore providing significant practical application for athletes. Indeed, following a single soccer match, athletes can experience reduced performance and significant muscle damage for up to 72 hours (Ascensão et al., 2008).
This information has been added in the manuscript in lines 360-394.
References
Ascensão, A., Rebelo, A., Oliveira, E., Marques, F., Pereira, L., & Magalhães, J. (2008). Biochemical impact of a soccer match – analysis of oxidative stress and muscle damage markers throughout recovery. Clinical Biochemistry, 41(10-11), 841-851.
Borrelli, F., Aviello, G., Romano, B., Orlando, P., Capasso, R., Maiello, F., et al. (2009). Cannabidiol, a safe and non-psychotropic ingredient of the marijuana plant Cannabis sativa, is protective in a murine model of colitis. Journal of Molecular Medicine, 87(11), 1111-1121.
Cochrane-Snyman, K. C., Cruz, C., Morales, J., & Coles, M. (2021). The effects of cannabidiol oil on noninvasive measures of muscle damage in men. Medicine and Science in Sports and Exercise, 53(7), 1460-1472.
Durst, R., Danenberg, H., Gallily, R., Mechoulam, R., Meir, K., Grad, E., et al. (2007). Cannabidiol, a nonpsychoactive Cannabis constituent, protects against myocardial ischemic reperfusion injury. American Journal of Physiology-Heart and Circulatory Physiology, 293(6), H3602-H3607.
Gallily, R., Yekhtin, Z., & Hanuš, L. O. (2018). The anti-inflammatory properties of terpenoids from cannabis. Cannabis and Cannabinoid Research, 3(1), 282-290.
Hatchett, A., Armstrong, K., Hughes, B., & Parr, B.B. (2020). The influence cannabidiol on delayed onset of muscle soreness. International Journal of Physical Education, Sports and Health, 7, 89-94.
Isenmann, E., Veit, S., Starke, L., Flenker, U., & Diel, P. (2021). Effects of cannabidiol supplementation on skeletal muscle regeneration after intensive resistance training. Nutrients, 13(9), 3028.
Kim, J., & So, W-Y. (2019). Effects of acute grape seed extract supplementation on muscle damage after eccentric exercise: A randomized, controlled clinical trial. Journal of Exercise, Science, and Fitness, 17(2), 77-79.
Malfait, A. M., Gallily, R., Sumariwalla, P. F., Malik, A. S., Andreakos, E., Mechoulam, R., & Feldmann, M. (2000). The nonpsychoactive cannabis constituent cannabidiol is an oral anti-arthritic therapeutic in murine collagen-induced arthritis. Proceedings of the National Academy of Sciences, 97(17), 9561-9566.
Mason, J., Rahlf, A. L., Groll, A., Wellmann, K., Junge, A., & Zech, A. (2021). The interval between matches significantly influences injury risk in field hockey. International Journal of Sports Medicine, 43(3), 262-268.
Mecha, M., Feliú, A., Iñigo, P. M., Mestre, L., Carrillo-Salinas, F. J., & Guaza, C. (2013). Cannabidiol provides long-lasting protection against the deleterious effects of inflammation in a viral model of multiple sclerosis: a role for A2A receptors. Neurobiology of Disease, 59, 141-150.
It may be appropriate to include "Acute supplementation" in the title.
This has been added to the title.
Considering adding the Hatchett (2020) paper to the discussion.
This article has been added to the discussion, in lines 373-375 (see response to first comment above) and in lines 546-547: “…and a significant benefit has been observed using full spectrum CBD and MCT oil.”

Round 2
Reviewer 3 Report
The authors have addressed most of my comments. Still needed is mention in the Discussion of the limitation of lack of dietary data. Diet is a variable that may have affected outcomes.
Author Response
Thank you for your comment. The following was added to lines 571-574: “Finally, diet was not controlled. It is well known that diet affects body composition and inflammation, thus potentially affecting the outcomes in this study. Participants in this study were instructed not to make any conscious changes to their regular dietary intake or use of dietary supplements during the intervention.”
